# Fluid Intake and the Occurrence of Erosive Tooth Wear in a Group of Healthy and Disabled Children from the Małopolska Region (Poland)

**DOI:** 10.3390/ijerph20054585

**Published:** 2023-03-04

**Authors:** Beata Piórecka, Małgorzata Jamka-Kasprzyk, Anna Niedźwiadek, Paweł Jagielski, Anna Jurczak

**Affiliations:** 1Department of Nutrition and Drug Research, Institute of Public Health, Faculty of Health Science, Jagiellonian University Medical College, Skawińska 8, 31-066 Krakow, Poland; 2Department of Paediatric Dentistry, Institute of Dentistry, Jagiellonian University Medical College, Montelupich 4, 31-155 Krakow, Poland

**Keywords:** fluid intake, erosive tooth wear, BEWE, healthy and with disabilities children

## Abstract

*Background:* The aim of this study was to analyse the relationship between the type and amount of fluid intake and the incidence of erosive tooth wear in a group of healthy children and children with disabilities. *Methods:* This study was conducted among children aged 6–17 years, patients of the Dental Clinic in Kraków. The research included 86 children: 44 healthy children and 42 children with disabilities. The prevalence of erosive tooth wear using the Basic Erosive Wear Examination (BEWE) index was assessed by the dentist, who also determined the prevalence of dry mouth using a mirror test. A qualitative-quantitative questionnaire on the frequency of consumption of specific liquids and foods related to the occurrence of erosive tooth wear, completed by the children’s parents, was used to assess dietary habits. *Results:* The occurrence of erosive tooth wear was determined for 26% of the total number of children studied, and these were mostly lesions of minor severity. The mean value of the sum of the BEWE index was significantly higher (*p* = 0.0003) in the group of children with disabilities. In contrast, the risk of erosive tooth wear was non-significantly higher in children with disabilities (31.0%) than in healthy children (20.5%). Dry mouth was significantly more frequently identified among children with disabilities (57.1%). Erosive tooth wear was also significantly more common (*p* = 0.02) in children whose parents declared the presence of eating disorders. Children with disabilities consumed flavoured water or water with added syrup/juice and fruit teas with significantly higher frequency, while there were no differences in quantitative fluid intake between groups. The frequency and quantity of drinking flavoured waters or water with added syrup/juice, sweetened carbonated, and non-carbonated drinks were associated with the occurrence of erosive tooth wear for all children studied. *Conclusions:* The group of studied children presents inappropriate drinking behaviours regarding the frequency and amount of beverages consumed, which, especially in a group of children with disabilities, may contribute to the formation of erosive cavities.

## 1. Introduction

The disabled population is a group at an increased risk of oral diseases. The reasons for this phenomenon can be found not only in the existence of numerous barriers to access to dental care but also in difficulties in implementing proper dietary and hygienic habits in this group of people. People with disabilities have limited access to health services, including routine treatment, which leads to non-disability-related health inequalities [1].

Difficulties with swallowing, eating, salivating, chewing, and unsatisfactory overall oral aesthetics may be present among people with Down syndrome, the most common genetic cause of intellectual disability. A higher prevalence of periodontal lesions has been identified in this group of individuals, which may be caused by the patient’s self-injury to oral tissues. A higher incidence of dental caries was also observed in the group of people with disabilities due to different craniofacial anatomy, functional disorders, or parafunctions. Children with physical and intellectual disabilities constitute a group that needs early and regular dental care in order to prevent and limit the severity of the pathologies observed [2,3,4].

According to the 2014 Polish Population Health Survey, disabled persons, by Polish criterion in the age group 0–14 years old, constituted 3.7% of the total. The data showed that the largest group of children with disabilities was recorded among 10–14-year-olds (5%), among 5–9-year-olds (4%), and less than 3% among the youngest children. More children with disabilities lived in urban areas than in rural ones, 140,000 vs. 72,000, respectively [5].

Dental erosion is the dissolution of dental hard tissues caused by acids of a non-bacterial origin. Erosive tooth wear is tooth wear with dental erosion as the primary etiological factor. As erosive tooth wear has serious long-term implications, it is important to establish its prevalence and its associated and aetiological factors [6]. The development of erosive tooth wear lesions may depend on internal factors, such as the state of health, the structure of the tooth, the structure and amount of saliva produced, as well as on external factors, mainly eating and drinking behaviour [7].

A systematic review presented that citrus fruits had a significant positive relationship with dental erosion. In addition, carbonated drinks and the consumption of acidic drinks at bedtime increased the risk of erosive tooth wear in adolescents. For sport/energy drinks and fruit juice, results were inconclusive [8].

Dental erosion has been considered an oral manifestation of eating disorders (i.e., anorexia, bulimia) associated with vomiting practices. The meta-analysis presented that patients with eating disorders and with risk behaviour of eating disorders had more risk of erosive tooth wear [9].

The literature suggests [10,11] that pathological conditions characterised by reduced salivary flow, i.e., salivary gland inflammation, Sjögren’s syndrome, or other symptoms, are factors that may influence the formation and development of dental erosion. The composition of saliva is particularly important in protecting against erosive processes, and normal salivary flow enables the dilution of acid concentrations of non-bacterial origin.

Dental erosion affected 42.3% of the participants in the young adult Polish population and 24.3% of the 15-year-old adolescent population [12,13]. To the best of the authors’ knowledge, the evaluation of factors influencing the development of erosive tooth wear among children with disabilities in Poland has not yet been conducted.

The aim of this study was to analyse the relationship between the type and amount of fluid intake and the incidence of erosive tooth wear in a group of healthy children and children with disabilities.

## 2. Materials and Methods

### 2.1. Study Design

This observational cross-sectional study was conducted between June and October 2019 among children of patients of a private dental practice in Kraków contracted by the National Health Fund for orthodontic treatment. A total of 101 questionnaires were collected, of which, after applying an exclusion criterion and verifying the completeness of the collected data, responses concerning 86 children were included in the evaluation, 44 healthy children and 42 children with disabilities, mainly Down syndrome (73.8%) and single cases of the following chronic conditions: childhood cerebral palsy, retinoblastoma, deletion syndrome, vertebrae damage, psychomotor retardation, body asymmetry, and motor aphasia. Inclusion criteria for this study: age 6–17 years and not taking medication affecting saliva secretion (inhaled medication used for bronchial asthma). Exclusion criteria for this study included: lack of parental consent, as well as lack of patient/child cooperation during the dental assessment.

All participants were informed about the conditions and procedure of this study and gave written consent to participate in the study. This study was conducted in accordance with the Declaration of Helsinki for medical research and received approval from the Bioethics Committee of Jagiellonian University (no. 1072.6120.138.2019 of 27 June 2019).

### 2.2. Data Collection

Parents/legal guardians of children were asked to answer a survey questionnaire related to dental treatment before their child entered the dental practice. In this study, no power calculation to estimate sample size was conducted. Dental observation of the occurrence and severity of erosive tooth wear and dry mouth was carried out in the case of children with disabilities by the orthodontics specialist Elżbieta Radwańska and in the group of healthy children by the dentist Barbara Noga. The calibration was not performed.

During the oral review, the children’s prevalence and severity of erosive tooth wear were assessed by noting the highest BEWE value for each sextant. On this basis, the child was categorised into a risk group based on the severity of erosive tooth wear and defined as 0–2—no risk (grade 1), 3–8—low risk (grade 2), 9–13—moderate risk (grade 3), and ≥14—high risk (grade 4) [14].

A mirror test was also performed to assess the presence of dry mouth. This index is based on a 3-point scale in the following categories: I no resistance (the mirror slides freely over the mucosa), II slight resistance (slight resistance is felt when moving the mirror), III significant resistance (the mirror sticks to the mucosa) [15].

To assess dietary behaviour, the authors used selected questions from the questionnaire on the frequency of consumption of specific products and liquids. These questions were modelled after the KomPAN Questionnaire developed by the Team of Behavioural Determinants of Nutrition, Committee on Human Nutrition Science, Polish Academy of Sciences (PAN) [16].

The questionnaire also included questions about selected socio-economic characteristics of the respondents, specific hygiene behaviours related to oral health maintenance, e.g., frequency of tooth brushing, and information about the general health of children, including subjective feelings of dry mouth. Parents/legal guardians were asked about the presence of medical conditions such as diabetes, asthma, Sjögren’s syndrome, xerostomia, inflammation of the salivary glands, and other conditions that increase the risk of erosive tooth wear, and whether the children were on continuous or regular medication (at least three times a week) and taking selected dietary supplements.

Parents were also asked to provide their child’s current height and weight, from which a body mass index (BMI, kg/m^2^) was calculated to assess the children’s nutritional status. The BMI values of each subject were related to national centile grids for age and sex, taking into account WHO criteria [17].

### 2.3. Statistical Analysis

Statistical analyses were performed using Statistica 13.0 PL. Due to the nature of the collected data, the evaluation of differences in responses tested with the χ2 test and the Mann-Whitney-U test as a non-parametric equivalent of the Student’s t-test was used. In the description of the results, group A denotes healthy children, while group B denotes children with disabilities.

Differences in respondents’ answers were checked for the presence of disability, age groups, dryness of the mouth (no dryness—level 1, presence of dryness—levels 2 and 3 in the classification of the mirror test), and for the risk of dental erosion according to the adopted interpretation of the BEWE index (group 1—no risk, group 2—low, moderate and high risk). The level of statistical significance was set at *p* < 0.05.

## 3. Results

### 3.1. Characteristics of Participants

The mean age of all children studied was 10.78 ± 2.96 years. There were no differences in the age of the respondents in the distinguished groups of healthy children and children with disabilities (Table 1). In the case of mothers of healthy children, only 18.2% reported not working, while 64.3% of mothers of children with disabilities did not work. In the study group of children with special needs, 35.7% lived in rural areas, while in the group of healthy children, significantly fewer rural residents (11.4%) were treated at a dental clinic (*p* = 0.0075).

None of the examined children had the following diseases associated with an increased risk of erosive tooth wear, i.e., diabetes, peptic ulcer disease, bronchial asthma, Sjögren’s syndrome, xerostomia, or inflammation of the salivary glands. On the other hand, in the group of children with disabilities, parents reported the occurrence of gastroesophageal reflux disease and eating disorders in children in single cases.

Statistically (*p* = 0.0002), significantly more (45%) parents of children with disabilities confirmed that their child was taking medications on a regular basis compared to healthy children (9%).

There was no statistically significant difference in the parents’ answers regarding dietary supplements taken by the child. 40% of parents gave their children supplements containing vitamin C, while 7% provided preparations containing iron. As many as 71% of all surveyed parents reported that their child received other supplements.

### 3.2. Prevalence and Severity of Erosive Tooth Wear and Dry Mouth in Dental Assessment

According to the BEWE classification of non-carious erosive cavities, 26% of the total number of children in this study had erosive tooth wear. A statistically significant difference was observed for the cumulative value of the BEWE index between the evaluated groups (*p* = 0.0003). In the group of healthy children, the mean value of the BEWE index was 1.39 (min = 0, max = 8, SD = 2.16), while the mean value of BEWE was higher for children with disabilities, amounting to 2.60 (min = 0, max = 7, SD = 1.98). There were no differences in the occurrence of erosive tooth wear depending on the sex, age, and BMI of the child.

The risk of erosive tooth wear was non-significantly more common in children with disabilities (31.0%) than in healthy children (20.5%). Erosive tooth wear was significantly more common (*p* = 0.02) in children whose parents declared the presence of eating disorders.

In both groups, as interpreted by BEWE, the lesions were of low severity, and therefore the risk of erosive tooth wear in the study group was low. The severity of the changes in the occurrence of erosive tooth wear in the study groups is shown in Figure 1.

For almost all healthy children (97.7%), no resistance was found when the dental mirror was moved along the cheek surface, i.e., they were properly hydrated. In contrast, for more than half of the children with disabilities (57.1%), slight resistance was found during the examination (Figure 2). This result was statistically significant (*p* < 0.0001). The survey questionnaire asked about the subjective feeling of dryness in the mouth. There was no statistically significant difference in the group of healthy and disabled children. Only 12% of parents of all the examined children reported dry mouth.

### 3.3. Oral Hygiene Behaviour in the Study Group of Children

In maintaining oral hygiene in the group of children with disabilities, parents used an electric toothbrush significantly more often, while healthy children used dental floss (*p* = 0.0032) and chewed sugarless gum after meals (*p* = 0.0373) significantly more often compared to children with disabilities (Table 2).

There are also significant differences regarding the frequency of children’s visits to the dental practice. For healthy children, 61.4% visit the dental practice every six months, while 42.9% of children with disabilities visit more often than every six months for dental check-ups (*p* = 0.0057).

### 3.4. Qualitative and Quantitative Fluid Intake in a Group of Healthy and Disabled Children

The frequency of consumption of specific beverages in the groups of healthy and disabled children is shown in Table 3, and the results indicate a significantly higher frequency of consumption of flavoured waters or waters with juice syrup and fruit tea in the group of children with disabilities compared to the control group. No differences were observed in the quantitative consumption of specific liquids and total fluid intake (TFI) in the study groups of children (Table 4).

### 3.5. Fluid Intake in a Study Group of Children and the Incidence of Erosive Tooth Wear

It was confirmed that erosive tooth wear changes were significantly more frequent in children consuming more sweetened carbonated and non-carbonated drinks and black tea, as well as drinking more liquids per day (Table 5). Also close to the accepted limit of statistical significance were flavoured waters or waters with added syrup/juice.

## 4. Discussion

In this study, the incidence of erosive tooth wear was 26% of the total number of children, and these were mostly lesions of minor severity. The exclusion criteria for the study group of children comprised the use of inhaled bronchial asthma medications. The cumulative assessment of the BEWE index showed a significant difference between the groups of children according to the presence of a disability, while in the interpretation of the BEWE, the risk of erosive tooth wear in a study group was non-significantly more frequent for children with disabilities (mainly with Down Syndrom) than for the healthy ones.

Similar results were obtained in a study conducted in Dubai in 2019, but dental erosion was significantly higher in children with Down Syndrome compared to healthy children (34% vs. 15.3%) [18].

Among the group of children with disabilities, as many as 57.1% showed slight resistance when moving a mirror in the mouth. This may be related to the amount of fluid consumed and the effect of medication, which, however, was not investigated in this study. A dry mouth can be one of the symptoms of dehydration. In the study group, besides the effect of frequency, the amount of fluid intake was also evaluated. Different studies show that children and adolescents in Europe do not drink enough water [19,20]. Decreased salivary flow causes a decrease in clearance rate, leading to an increase in the risk of erosive tooth wear, especially in the case of physical activity [21].

In the survey, we confirmed that erosive tooth wear was significantly more frequent in children consuming more sweetened carbonated and non-carbonated drinks and black tea, as well as drinking more fluid per day. Also close to the accepted limit of statistical significance were flavoured or syrup/juice-infused waters, the amount of consumption of which may influence the development of erosive tooth wear, which is related to the low pH of these drinks (pH < 4.5). A higher frequency of consumption of flavoured waters or waters with juice syrup and fruit tea was observed in the group of children with disabilities.

The findings of the present study are in accordance with the results of a systematic review, where carbonated drinks were significantly positively associated with dental erosion in adolescents [8].

Also, a positive correlation was observed between the erosive lesions of the anterior teeth and the frequency of consumption of carbonated and energy drinks in the population of adolescents aged 15 in Poland [12].

However, in the population of 18-year-old young adults in Poland, drinking behaviour, like frequent consumption of fruit teas and energizing beverages, was connected with dental erosion. Also, hygienic habits, medical conditions such as asthma, eating disorders, and oesophageal reflux showed statistical significance associated with erosive tooth wear [13].

Children and adolescents from Poland make mistakes regarding the frequency of beverage consumption. The vast majority of schoolchildren from Kraków and the surrounding area indicate that they consume water daily about three times a day, but more than a third of them choose flavoured water or water with added juices/syrups [22].

A national study by Jessa J. and Hozyasz K. [23] indicated that children aged 6 months to 18 years hospitalised in Warsaw in 2016 at the Department of Paediatrics of the Mother and Child Institute were significantly more likely to drink flavoured waters.

In contrast, in a group of adolescents from the region of Podkarpacie (Poland), sugar beverages (soft drinks) were consumed most frequently, and respondents chose energy drinks more often than isotonic beverages. All the beverages indicated have an adverse effect on the development of erosive tooth wear [24].

Similarly, as in the presented study of a group of children from the Małopolska region, a study by Alves et al. showed an association of dental erosion with the consumption of soft drinks (including sweet carbonated and non-carbonated drinks), but also fruit juices [25].

In a cross-sectional study on a sample of 400 children from Valencia (Spain), a positive correlation was observed between the presence of tooth erosion and frequent consumption of fruit juices, fizzy drinks, and isotonic drinks (*p* < 0.05), showing a higher correlation if the liquid was held in the mouth before swallowing [26].

The study among adolescents in Stockholm County [27] diagnosed that erosive lesions were significantly correlated with soft drink consumption, the use of juice or sport drinks as a thirst quencher after exercise, and tooth hypersensitivity when eating and drinking.

The presented studies lack uniform nomenclature of individual types of beverages, and they do not specify the type of fruit juice, which makes it difficult to compare the results.

It is recognized that beverages with high calcium content, like milk or calcium-enriched juices, may reduce the risk of dental erosion. Therefore, adequate consumption of milk and dairy products is important in the prevention of dental erosion [8].

In the study by Guelinckx et al. [28], data from 3611 children and 8109 adolescents were retrieved from 13 countries, including Poland. In the total sample, the highest mean intakes were observed for water (738 ± 567 mL/day), followed by milk (212 ± 209 mL/day), regular soft beverages (RSB) (168 ± 290 mL/day), and juices (128 ± 228 mL/day). Large contributions of hot beverages, like black or fruit tea, to total fluid intake (TFI) were reported in the total children sample of Poland, which is culturally conditioned.

In the study group of children from the Małopolska region, the amount of milk consumption was similar (median 200 mL/day).

In a study by Hasselkvist et al., the development of erosive tooth wear was influenced by lesser sour milk intake and more frequent intake of drinks between meals [29].

Besides the consumption of acidic drinks, a lifestyle that may be conducive to such consumption, such as sedentary living, excessive screen viewing activities, as well as being overweight, may contribute to the development of erosive wear [8]. Numerous studies showed a positive correlation between the frequency and quantity consumption of sugar-sweetened beverages and body mass index in children and adolescents [30].

In this study of children from the Małopolska region, there was no statistically significant difference in the occurrence of erosive tooth wear in relation to the sex, age, and BMI of the child. However, the declared amount of acidic liquid consumption was associated with the occurrence of erosive tooth wear.

In study by Tschammler et al., a total of 223 children aged 4–17 years children with obesity and extreme obesity compared to children with normal weight had significantly higher erosive wear and caries of deciduous and permanent teeth [31].

People with intellectual disability (ID) are characterised by a high prevalence of incorrect eating patterns, as well as a high risk of becoming overweight or obese. The results of this study from Poland showed that excess body weight was observed in 66.7% and obesity in 38.9% of the respondents (seven subjects) with ID [32].

Due to the lack of Polish data regarding the quality of fluid consumption of children and adolescents with ID, it is impossible to confront the results of this study with the findings of other Polish authors.

Oral hygiene is also an important protective factor in the prevention of erosive cavities. Children with disabilities have difficulty maintaining proper oral hygiene. In maintaining oral hygiene in the surveyed group of children with disabilities, parents significantly more often use an electric toothbrush, while healthy children significantly more often use dental floss and chew sugar-free gum after meals compared to children with disabilities. The surveyed group of parents of children with disabilities is aware of the importance of oral health in relation to the health of their child due to the declared high frequency of visiting the dental clinic with their child. Almost half of the parents/guardians of the studied children had a university degree.

Dental treatment of disabled people in Poland is provided free of charge. In addition to the services guaranteed in the Polish system of health care for disabled people, they have access to treatment with the best materials and treatment methods. People with disabilities are reimbursed by the state for treatment under general anesthesia [33].

The data obtained from parents/guardians of disabled and/or chronically ill children living in Poznań and Białystok (Poland) showed that up to 18.5% of children with disabilities had never been to a dentist. The most common reasons for a dental visit were changes within a tooth noticed by a parent (25.5%) or a dental check-up (25%). Only 67.5% of respondents reported no access barriers to dental treatment [34].

### Strengths and Limitations of this Study

In the presented study, the limitation of the interpretation of the results may be due to the specificity of the collection of material and the small size of the study group. Other limitations are the lack of a power calculation to estimate the sample size and also the lack of results regarding the assessment of the prevalence of caries and oral hygiene. Among the strengths of the study, it should be emphasised that few studies provide a quantitative assessment of the fluids drunk by a group of children with special needs.

## 5. Conclusions

The group of children studied presents inappropriate drinking behaviours regarding the quality of the beverages consumed, which, especially in a group of children with disabilities, may contribute to the formation of erosive cavities. Disabled children cannot perform hygiene procedures and make decisions in the field of eating habits on their own. Therefore, parental education on the relationship of food and fluid intake to oral hygiene and general health in the future should be increased.

## Figures and Tables

**Figure 1 ijerph-20-04585-f001:**
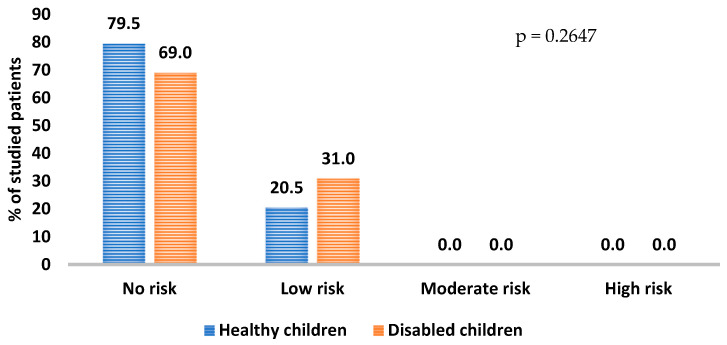
Interpretation of the severity of erosive tooth wear in selected groups of children.

**Figure 2 ijerph-20-04585-f002:**
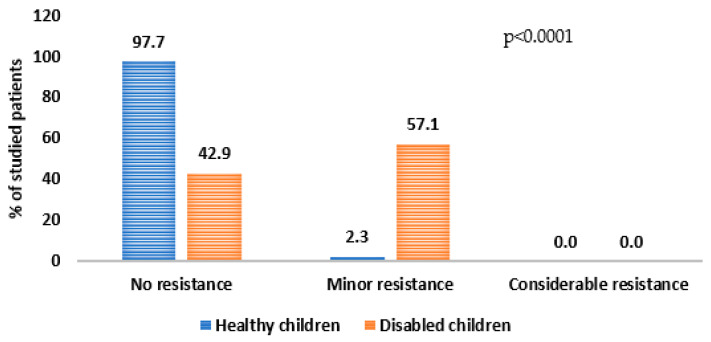
Interpretation of mirror test results in selected groups of children.

**Table 1 ijerph-20-04585-t001:** Socio-demographic characteristics and nutritional status of children, including healthy and disabled groups.

Variable	Category	Healthy Children (*n* = 44)	Disabled Children (*n* = 42)	*p*-Value
Age	6–11 12–17	61.4 38.6	64.3 35.7	0.7793
Gender	Girls	63.6	59.5	0.6950
	Boys	36.4	40.5	
Place of residence	Urban area	88.6	64.3	0.0075
	Rural area	11.4	35.7	
Mother’s education	Primary	0.0	2.4	0.5943
	Basic vocational	4.5	21.4	
	Secondary	27.3	21.4	
	Higher Vocational	22.7	7.1	
	Master’s Degree	45.5	47.6	
Father’s education	Primary	0.0	9.5	0.3288
	Basic vocational	6.8	9.5	
	Secondary	25.0	26.2	
	Higher Vocational	20.5	14.3	
	Master’s Degree	47.7	40.5	
How do you assess your material situation?	Bad	0.0	0.0	0.1876
	Satisfactory	2.3	14.3	
	Sufficient	25.0	31.0	
	Good	56.8	40.5	
	Very Good	15.9	14.3	
Does the mother work?	No	18.2	64.3	<0.0001
	Yes	81.8	35.7	
Children BMI result	Underweight	4.5	14.3	0.8002
	Within norm	75.0	61.9	
	Overweight	20.5	14.3	
	Obese	0.0	9.5	

Tests: χ^2^ test.

**Table 2 ijerph-20-04585-t002:** Oral hygiene behaviours of healthy and disabled children.

Variable	Category	Healthy Children (*n* = 44)	Disabled Children (*n* = 42)	*p*-Value
Electric toothbrush	No	56.8	33.3	0.0288
	Yes	43.2	66.7	
Interdental toothbrushes	No	95.5	97.6	0.5845
	Yes	4.5	2.4	
Manual toothbrush	No	27.3	45.2	0.0828
	Yes	72.7	54.8	
Dental floss	No	63.6	90.5	0.0032
	Yes	36.4	9.5	
Mouthwash	No	61.4	45.2	0.2182
	Yes	38.6	54.8	
Irrigator	No	97.7	90.5	0.9734
	Yes	2.3	9.5	

Tests: χ2 test.

**Table 3 ijerph-20-04585-t003:** Frequency of consumption (% of answers) of specific beverages, taking into account healthy and disabled children.

Fluids	Group	Never	Less Than Once a Week	Once a Week	a Few Times a Week	Every Day	*p*-Value
Water	A B	2.3 2.4	0.0 0.0	2.3 0.0	11.4 9.5	84.1 88.1	0.5852
Flavoured waters or with added syrup/juice	A B	40.9 26.2	18.2 11.9	15.9 16.7	18.2 26.2	6.8 19.0	0.0337
100% fruit/vegetable juices	A B	27.3 21.4	11.4 14.3	20.5 11.9	31.8 40.5	9.1 11.9	0.3925
Sweetened non-carbonated beverages	A B	59.1 61.9	20.5 26.2	15.9 7.1	4.5 4.8	0.0 0.0	0.6283
Sweetened carbonated beverages (e.g., cola drinks)	A B	47.7 52.4	25.0 31.0	20.5 7.1	6.8 7.1	0.0 2.4	0.5271
Black tea	A B	36.4 42.9	18.2 9.5	4.5 11.9	20.5 19.0	20.5 16.7	0.6237
Fruit tea	A B	47.7 31.0	20.5 16.7	11.4 14.3	13.6 23.8	6.8 14.3	0.0468
Milk and/or milk beverages	A B	4.5 19.0	13.6 11.9	18.2 7.1	27.3 35.7	36.4 26.2	0.2485
Isotonic beverages	A B	90.9 95.2	9.1 4.8	0.0 0.0	0.0 0.0	0.0 0.0	0.4336

A—healthy children, B—disabled children; Tests: Mann-Whitney U test.

**Table 4 ijerph-20-04585-t004:** Declared amount of beverages consumed [ml] per day, taking into account healthy and disabled children.

Fluids/Beverages [mL]	Healthy Children (*n* = 44)			Disabled Children (*n* = 42)			*p*-Value
	**Median**	**P25**	**P75**	**Median**	**P25**	**P75**	
Water	800	600	1100	600	400	1000	0.1792
Flavoured waters or with added syrup/juice	200	0	300	200	0	400	0.1543
100% fruit/vegetable juices	200	0	200	200	0	400	0.0691
Sweetened non-carbonated beverages	0	0	200	0	0	0	0.1799
Sweetened carbonated beverages (e.g., cola drinks)	0	0	200	0	0	200	0.3297
Black tea	200	0	200	0	0	200	0.3241
Fruit tea	0	0	200	200	0	200	0.4905
Milk and/or milk beverages	200	200	400	200	200	400	0.6196
Isotonic beverages	0	0	0	0	0	0	0.4870
Other	0	0	0	0	0	0	0.1166
Total fluids	1800	1500	2400	1900	1600	2600	0.7945

Tests: Mann-Whitney U test.

**Table 5 ijerph-20-04585-t005:** Declared amount of drinks consumed [ml] per day in relation to the assessment of the severity of erosive tooth wear according to BEWE for the total subjects.

Fluids/Beverages [mL]	No Risk (*n* = 64)			Low Risk (*n* = 22)			*p*-Value
	Median	P25	P75	Median	P25	P75	
Water	800	500	1000	800	400	1000	0.8568
Flavoured waters or with added syrup/juice	200	0	200	200	200	400	0.0590
100% fruit/vegetable juices	200	0	200	200	200	400	0.1275
Sweetened non-carbonated beverages	0	0	0	100	0	200	0.0079
Sweetened carbonated beverages (e.g., cola drinks)	0	0	200	200	0	200	0.0133
Black tea	0	0	200	200	0	200	0.0102
Fruit tea	0	0	200	200	0	200	0.4611
Milk and/or milk beverages	200	200	400	200	200	400	0.4240
Isotonic beverages	0	0	0	0	0	0	0.3924
Other	0	0	0	0	0	0	0.3873
Total fluids	1800	1500	2200	2200	1800	2800	0.0290

Tests: Mann-Whitney U test.

## Data Availability

Not applicable.

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
