# Peer review of "Fluid Intake and the Occurrence of Erosive Tooth Wear in a Group of Healthy and Disabled Children from the Małopolska Region (Poland)"

_ijerph, 2023, doi:10.3390/ijerph20054585_

Round 1
Reviewer 1 Report
Introduction section:
- The authors need to improve this item cited results observed by systematic reviews and meta analysis of related to dental erosion and the different methods for preventing and treating this problem.
Materials and Methods section:
- Please add the registration number.
- Also please can u illustrate the methods for evaluation in more details.
- Please add how did you calculate the sample size.
Result section:
- Well executed.
Discussion section:
- The reviewer think that discussion section must be improved regarding the agreed or conflicting results and also the clarification of results.
Conclusion section:
- Please add few statements into the conclusion section as it should be more descriptive. Also add the future scope of the study.
Author Response
Dear Reviewer,
thank you very much for the review and valuable remarks for improving the manuscript.
Point 1: The authors need to improve this item cited results observed by systematic reviews and meta-analysis of related to dental erosion and the different methods for preventing and treating this problem.
Response 1: Added in the introduction section
Point 2: Please add the registration number.
Response 2: It was presented in the content of the work. “The study received an approval of the Bioethics Committee of the Jagiellonian University (no. 1072.6120.138.2019 of 27 June 2019)”.
Point 3: Also please can u illustrate the methods for evaluation in more details.
Response 3: Added in the materials and methods section
Point 4: Please add how did you calculate the sample size.
Response 4: This is not a population study and no sample size was calculated
Point 5: The reviewer think that discussion section must be improved regarding the agreed or conflicting results and also the clarification of results.
Response 5: Improved discussion presentation
Point 6: Please add few statements into the conclusion section as it should be more descriptive. Also add the future scope of the study.
Response 6: Supplemented in the conclusion section
Reviewer 2 Report
This is an interesting manuscript dealing with an important subject. However, there are several issues that the authors need to address.
Introduction
Line 47. It is unclear why you write about Downs syndrome. Is that the most common disability? Please include the most common disabilities among children.
Line 61. Structure and amount of saliva. Is not composition of saliva a better description?
Line 62. Eating behavior- please add drinking behavior.
Material and methods:
Line 85. Please describe the questionnaire. Please also explain how the parents were selected. So of the 86 parents it was 42% that had children with disabilities- it seems very high.
Why did you not register caries status?
Line 105. Please describe the risk group categorization of erosion.
Line 105. Why did you not ask the child/parent about dry mouth experience and evaluate the presence of saliva and clinical signs of reduced salivary secretion?
Line 110. Please clarify the number of questions from the questionnaire you selected, what the questions were about and the answering alternatives.
Line 115. Were the parents not asked about their child’s fluoride use?
Line 128. U Mann Whitney test- should it not be Mann Whitney U test?
Results
Line 140. How can you know that the fact that the mother did not work was because of the need of the child?
In the MM section you write that you asked the parents about the general health of the child, and medication. However, you do not present those results.
Line 141-143. Please clarify this with children living in rural and treatment at a dental clinic. Is the accessibility to dental clinics lower in non-rural areas?
Line 156. Eating disorders- it is not clear in the MM section how this information was gained and what was classified as eating disorder?
Table 1. It would have been interesting to know if you found differences between children aged 6-11 years and 12-17 years regarding erosion severity and also if you found differences between girls and boys
Line 154. Here you write about risk of dental erosion is this only according to the results of the BEWE categorization? If so, should you not also take into account oral hygiene, fluoride exposure and salivary quantity and quality?
Figure 1
“little risk”- low risk?, “large risk” - high risk?
Table 2. Electric toothbrush- which brands were used. There are many different brands on the market and all might not be efficient plaque removers.
Table 3. what was the percentage of sugars in those beverages?
Table 5. Was it black tea with sugar added?
Discussion
It would be nice if information about dental routines for children with disabilities in Poland is described. I.e does check-ups cost, can the get preventive dental care free of charge?
Line 226. Please clarify how this section is correlated to the results of the present study?
Line 236. Please clarify how this section is correlated to the results of the present study?
Line 244. Please clarify how this section is correlated to the results of the present study?
Line 274. Please clarify how this section is correlated to the results of the present study?
Line 286. Please clarify how this section is correlated to the results of the present study?
Line 307. What is abnormal eating patterns?
Please discuss why different persons collected data for the healthy children and the children with disabilities.
Please also discuss the mirror test- is it reliable? Could another test be used?
Were the children/parents given any advice to decrease the risk of dental erosion?
Please discuss how you arrived at the number of children to include in each group.
Author Response
Dear Reviewer,
thank you very much for the review and detailed remarks for improving the manuscript.
Authors
